# Quick and Easy Covalent Grafting of Sulfonated Dyes to CMC: From Synthesis to Colorimetric Sensing Applications

**DOI:** 10.3390/polym14194061

**Published:** 2022-09-27

**Authors:** Lisa Rita Magnaghi, Camilla Zanoni, Denise Bellotti, Giancarla Alberti, Paolo Quadrelli, Raffaela Biesuz

**Affiliations:** 1Department of Chemistry, University of Pavia, Via Taramelli 12, 27100 Pavia, Italy; 2Unità di Ricerca di Pavia, Consorzio Interuniversitario Nazionale per la Scienza e Tecnologia dei Materiali (INSTM), Via G. Giusti 9, 50121 Firenze, Italy; 3Department of Environmental and Prevention Sciences, University of Ferrara, 44121 Ferrara, Italy; 4Faculty of Chemistry, University of Wrocław, 50-383 Wrocław, Poland

**Keywords:** biocompatible materials, CMC, covalent grafting, sulfonated dyes, design of experiments, colorimetric sensors

## Abstract

Carboxymethyl cellulose, the most promising cellulose-derivatives, pulls together low cost, abundancy, biocompatibility, unique properties and, unlike the precursor, chemical reactivity. This latter aspect arouses the curiosity of chemists around the possibility of chemical modification and the production of interesting functional materials. Here, a two-step reaction is proposed for the covalent anchoring of a wide variety of molecules containing sulfonic groups to CMC. The strength points of the proposed pathway have to be found in the quick and easy reactions and workup that allow to obtain ready-to-use functional materials with very high yields. Having in this case exploited a pH-sensitive dye as a sulfonated molecule, the functional material is an interesting candidate for the development of colorimetric miniaturized sensors via the following drop-casting deposition: once optimized sensors preparation by design of experiments, an example of application on real samples is reported.

## 1. Introduction

In recent years, the environmental concerns surrounding conventional petroleum-based materials have stimulated the research on natural macromolecules, which guarantee biodegradability and sustainability [1]. Among all, cellulose-based derivatives have deservedly gained a prominent role as natural, biodegradable, renewable, versatile and economical materials [2]. It must be underlined that the large availability of cellulose, the most abundant biopolymer in nature, and the countless opportunities to produce industrial-appealing derivatives from renewable or even waste sources, played a crucial role in the widespread diffusion of this type of substrate [2].

Cellulose offers a wide variety of natural derivatives, and we focused our attention on carboxymethylcellulose (CMC), an anionic and water-soluble derivative in which some of the hydroxyl groups of the glucopyranose units are substituted by sodium carboxymethyl groups (−CH2COONa) [3,4]. CMC was first synthesized in 1918; however, the commercial production of these all-important polymeric materials was first depicted in Germany in the early 1920s [4] and nowadays is one of the most common industrially available cellulose ethers, with a worldwide production of 300,000 ton/year [3]. 

Moving to a more chemical point of view, CMC guarantees appealing properties such as biodegradability, biocompatibility, improved solubility in aqueous or organic solvents and the presence of reactive functions, which in addition can be tuned at will, varying CMC features such as the degree of substitution or the molecular weight [3]. Thanks to its unique features, this material has been widely exploited in different application fields, ranging from textiles to the food industry and cosmetics, from pharmaceuticals to biomedical [3,4,5].

Unlike the parent cellulose, the presence of two kinds of reactive sites (carboxyl and hydroxyl) on the CMC backbone paved the way to a variety of straightforward further chemical modifications, intended to further tune CMC properties for desired application [3]. In the last decade, several synthetic approaches have been tested for CMC chemical modification, perfectly summarized by Pettignano and coworkers in a recent review [3]. Focusing on the carboxyl moiety modifications, the following three main strategies emerge from the literature: amidation, esterification and multicomponent reactions [3,6], while, in the case of hydroxyl moiety derivatization, sulfonation was the most common approach [7,8].

Trying to piece together our background on chemical modifications of polymeric substrates and the synthetic pathways proposed in the literature for CMC, of which the previous paragraph is only a short summary, we developed a 2-step synthetic pathway for the covalent grafting to CMC of molecules containing sulfonic groups. A similar methodology has been already proposed by our research group for the covalent binding of sulfonated dyes to ethylene vinyl alcohol copolymers (EVOH) [9,10,11,12,13], but in the present work, the entire procedure has been revised and adapted for the biopolymeric material. The first driving force of this work was the simplicity of both the synthetic pathway and workup procedure as follows: as a matter of fact, the entire process was completed in a few hours, achieving very high yields and obtaining reproducibly versatile functionalized materials. The second one was the applicability of the obtained functionalized CMC for sensing applications. For this reason, after a full characterization of the new material, the optimization of colorimetric pH-sensitive sensors’ deposition via drop-casting is presented. Finally, as a “proof of concept”, their application as freshness sensors for protein foods, which is one of the most famous fields of application of pH-sensitive dyes in the last years, is discussed [14,15,16,17].

What clearly distinguishes our approach from other equally interesting proposals in the literature is the prominent role attributed to CMC and the easy and tunable procedure for sensor preparation. On one hand, while CMC is commonly used as an additive or minor component, in our case it is exploited as a solid support for the reactive dye thanks to the covalent functionalization pathway developed. On the other hand, the versatile nature of biocompatible materials allows for optimization of the sensors depending on the specific application required, as demonstrated in this case for food freshness detection.

## 2. Materials and Methods

*o*-Cresol red Analytical Reagents grade, thionyl chloride solution 1M in dichloromethane, sodium hydroxide pellets, potassium hydroxide pellets, potassium hydrogen phthalate (99.9% purity), potassium chloride, nitric acid, hydrochloric acid solution 1M, glacial acetic acid, toluene, DMF and dichloromethane (DCM) were purchased by Merck. The carbonate-free stock solutions of 0.1M KOH were prepared by diluting concentrated KOH and then potentiometrically standardized with the primary standard potassium hydrogen phthalate. Carboxymethyl cellulose (CarboCell MM250, DS 0.80–1.0, average Mw~250,000, viscosity 2% solution 1000–3000 cP) was provided by Lamberti Spa (Gallarate—21013 VA, Italy) and cellulose pulp sheets were provided by Barbè Srl (Mortara—27036 PV, Italy). Food samples were bought in a local supermarket (UNES Supermarkets, via Fratelli Cervi, 11 27100 Pavia, Italy), on the same day as supplier’s delivery.

### 2.1. Development of Synthetic Pathway for Covalent Anchoring of Sulfonated Dyes to CMC

The synthetic pathway for covalent anchoring of sulfonated dyes to carboxymethylcellulose (CMC) consisted of the following two reaction steps: (i) dye activation through chlorination and (ii) biopolymer functionalization by nucleophilic substitution, similarly to the procedure proposed in the literature for covalent anchoring on EVOH copolymers containing hydroxyl groups [9,10,11,12,13,18]. Firstly, the general procedure is reported while, in the following section, the procedure optimization is discussed. The covalent anchoring to CMC was also performed for different sulfonated and carboxylic dyes observing no significant difference changing the dye but, for brevity’s sake, only the results regarding *o*-cresol red will be discussed.

The first step was performed dissolving a given amount of CR, ranging from 0.3 to 0.05 mmol, in 10 mL SOCl_2_, heating at reflux for the defined time and evaporating the excess SOCl_2_. Then, 5 g CMC was suspended in the reaction solvent under stirring and heated at 65 °C; when the temperature was reached, a freshly prepared sulfonyl chloride solution in the reaction solvent (10 mL) was added dropwise to the polymer solution. After 3 h, the reaction mixture was cooled at room temperature and in an ice bath. Finally, the functionalized CMC was filtered under vacuum, washed two times with 30 mL aliquots of DCM to remove the unreacted dye excess and left to dry overnight.

Referring to the general procedure described above, different reaction solvents and dye amounts were tested to develop the final synthetic pathway. Being these parameters completely independent one another and resulting in different characteristics of the final material, an OVAT (one variable at time) approach was followed in the conditions definition. As for reaction solvent, DMF, DCM and toluene were tested and the precipitation yield was calculated to select the best candidate; in the case of starting dye amount, a wide range of values was investigated, moving from 0.30 to 0.05 mmol and the color intensity of the obtained CR-CMC@ was measured to calculate the stoichiometric amount of dye.

### 2.2. CR-CMC@ Physico-Chemical and Optical Characterisation

Differential scanning calorimetry (DSC) analyzes were performed by heating the samples (~5 mg) from −20 to 250 °C at a rate of 5 °C/min under an N_2_ atmosphere in Al crucibles by a Q2000 instrument interfaced with a TA 5000 data station (TA Instruments). The heating process was preceded by an isothermal to remove the sorbed humidity. 

Fourier transform infrared (FT-IR) spectra were recorded using a Nicolet (Madison, WI, USA) FT-IR iS10 spectrometer equipped with an attenuated total reflectance (ATR) sampling accessory (Smart iTR with a diamond plate) by co-adding 32 scans in the range from 4000 to 650 cm^−1^ with the resolution set at 4 cm^−1^.

^1^H-NMR spectra of CR, CMC, CR-CMC@ and CR and CMC mixture were acquired using a Bruker Avance Neo 400 MHz spectrometers dissolving the samples in D_2_O.

UV-Vis spectra of CMC, CR-CMC@ solutions 0.5% (*w*/*v*) in 0.1 M HNO_3_, phosphate buffer at pH 7.00 and 0.1 M NaOH were acquired, using the Jasco V-750 spectrophotometer and compared with the corresponding spectra of the dye dissolved in the same media (~10 μM).

### 2.3. Determination of Acid-Base Constants for CR, CMC, CR-CMC@

Protonation constants for CR and CMC were calculated from potentiometric and pH-dependent UV-Visible titration curves registered at *T* = 298 K and ionic strength *I* = 0.1 M (KCl). The potentiometric apparatus consisted of an Orion EA 940 pH-meter system provided with a Metrohm 6.0234.100, glass-body, micro combination pH electrode and a dosing system Hamilton MICROLAB 500, equipped with a 0.5 mL microburette. The thermostated glass cell was equipped with a magnetic stirring system, a microburet delivery tube and an inlet-outlet tube for nitrogen. High purity grade gas nitrogen was gently blown over the test solution in order to maintain an inert atmosphere. A constant-speed magnetic stirring was applied throughout. Solutions were titrated with carbonate-free KOH. The electrode was daily calibrated for hydrogen ion concentration by titrating HNO_3_ with alkaline solution under the same experimental conditions as above. The standard potential and the slope of the electrode couple were computed by means of SUPERQUAD [19] and Glee [20] programs. The purities and the exact concentrations of the ligand solutions were determined by the Gran method [21]. For the potentiometric titrations, aqueous solutions of CR = 1 − 1.5 × 10^−3^ M and *I* = 0.1 M (KCl), and CMC = 0.25% (*w*/*v*) and *I* = 0.1 M (KCl) were employed. 

Spectrophotometric pH-dependent titrations of CR and CR-CMC@ solutions were carried out in the pH range of 2.5–10.5, by following the changes in the most intense absorption bands in the visible region. The absorption spectra were recorded on a Varian Cary50 Probe spectrophotometer, in the range 250–800 nm, using a quartz cuvette with an optical path of 1 cm. For the spectrophotometric measurements, aqueous solutions of CR = 1 × 10^−5^ M and *I* = 0.1 M (KCl), and CR-CMC@ 0.25–0.5% (*w*/*v*) and *I* = 0.1 M (KCl) were prepared. In total, 3 mL of the sample was manually titrated directly in a quartz cuvette by addition of known volumes of KOH 0.1 M. The pH was monitored using a glass electrode daily calibrated for hydrogen ion concentration, as described above for potentiometric measurements. After each addition of KOH, the UV-Vis absorption spectrum of the sample solution was recorded.

All sample solutions employed in titrimetric and spectrophotometric methods were formulated with freshly prepared Milli-Q^®^ water. The HCl and HNO_3_ stock solutions were prepared by diluting concentrated HCl and HNO_3_ and then standardized with standard solution of KOH. The ionic strength was adjusted to 0.1 M by adding KCl. A glassware was employed throughout.

Data obtained by potentiometric method were processed with the HYPERQUAD [22] software, while spectroscopic data were refined using HYPSPEC [22] software. Both methods were employed to calculate the overall (β) and step (K) protonation constants of each studied system, which is considered a generic weak acid. The computed standard deviations (referring to random errors only) were given by the program itself and are shown in parentheses as uncertainties on the last significant figure. The distribution diagrams were computed using the HYSS [23] program.

### 2.4. CR-CMC@ Drop Deposition on Pure Cellulose: General Procedure and Optimization

CR-CMC@ aqueous solutions for drop deposition were prepared by pouring CR-CMC@ powder and glycerol in 10 mL, under heating and stirring till complete dissolution. Then a defined volume of solution was collected by a positive displacement pipette, dropped on cellulose and dried at RT for 2 h before the application

This general procedure was optimized by Full Factorial Design 2^3^ as follows: the variables under investigation were CR-CMC@ concentration, glycerol/CR-CMC@ ratio and drop volume; the levels defined for each variable are reported in Table 1. The canonical model equation, including the linear terms (3) and 2-terms interactions (3) was postulated and the training sample, whose experimental conditions are reported in row 1–8 in Appendix A were prepared to build the model. After that, three replicates of the centre point [0 0 0], described in row 9 in Appendix A were exploited for model validation.

The detection kinetic of weak acid analytes originating from acetic acid solution was exploited as a response for the design. To reliably compare the results obtained with different sensors, detection kinetics were measured at the following fixed conditions: 100 mL of AcOH solution 0.01 M were poured into a sealed box (V = 1.75 L) and the sensors, previously equilibrated at the alkaline form, were fixed to the top of the box by the means of a 3D-printed support, shown in Appendix A. It must be underlined that, to enhance sensors’ sensitivity towards acidic analytes and to increase the detection rate, the stoichiometric amount of strong base must be added, avoiding any excess to be sorbed by the sensor or the support. This stoichiometric amount was separately evaluated for each sensor and is reported in the last column of Appendix A.

To monitor detection kinetic, pictures of the sensors were acquired for 5 h till complete color conversion, following the picture acquisition procedure described in Appendix A, and HSL values were extracted using GIMP software [24]. The following two different responses were analyzed: (i) the time required for a complete conversion and (ii) the color intensity. As for the first response, the color conversion was judged as complete when the difference between H value, the hue component of the HSL space color, at given time and the final one was below 10% while, as for the second response, the S value at the equilibration time was used to measure color intensity. 

For each response, model coefficients were calculated, the model was validated comparing the predicted response value at [0 0 0] with the experimental one and, finally, a Pareto-optimal front was exploited to select the best composition.

Design of experiments was performed using the open-source software CAT [25].

### 2.5. CR-CMC@/Cell Sensor Physico-Chemical Characterisation

CR-CMC@/Cell sensors were first characterized registering FT-IR spectra of the sensors and sensors’ components, according to the procedures previously described. 

SEM images at different magnifications were acquired by EVO MA10 scanning electron microscope (SEM); the films were supported on graphite biadhesives fixed on Al stubs and subsequently transferred into the SEM chamber.

### 2.6. CR-CMC@/Cell Sensors Application: A “Proof-of-Concept”

Finally, CR-CMC@/Cell sensors were tested for the detection of weak acids released by spoiling protein foods, as it is widely known in the literature, [9,10,11,12,13,26,27,28] exploiting the same experimental setup described for synthetic vapor-generating solutions. Different cuts of chicken and turkey meat were tested, ranging from breast slices to minced meats and the sensors’ color evolution was registered while storing the foods in a domestic fridge (4 °C) for 5 days.

## 3. Results and Discussion

### 3.1. Development of CR-CMC@ Synthetic Pathway

Covalent anchoring of sulfonated dyes to polymeric materials containing hydroxyl groups had been already proposed in the literature, [9,10,11,12,13,18] and the general reaction scheme, reported in Figure 1 in the case of *o*-cresol red, was kept unchanged but, moving to biopolymers and in this case to CMC, the starting material presented completely different physical and chemical characteristics, such as granulometry, solubility, reactivity and degree of substitution, which necessitate different reaction conditions. 

Firstly, DMF, DCM and toluene were tested as reaction solvents with the double final aim of successfully anchoring the dyes to CMC and improving the precipitation yield. It must be underlined that in all the cases, covalent anchoring was performed at 65 °C without solubilization of CMC, which actually remained as powder under stirring during the entire reaction. To further increase the precipitation yield, the reaction mixture was cooled at RT and in an ice bath before filtering under vacuum CR-CMC@ and, in the case of toluene, the solvent volume was also reduced by half by evaporation. In Figure 2, CR-CMC@ obtained with different reaction solvents are shown, and in Table 2, the average precipitation yields are reported with standard deviations in brackets.

Precipitation yields were generally high; the best results were achieved using DCM and toluene after evaporation, but the latter was preferred due to the higher color intensity and the quicker filtration procedure.

Secondly, the stoichiometric amount of dye needed to be experimentally defined as follows: unfortunately, this amount could not be calculated since both the degree of substitution (DS) and the molecular mass (MM) were given as an average value by the producer and, by performing the anchoring directly on CMC powder, not all the reactive sites might be available. For this reason, the covalent anchoring was performed with decreasing amounts of dye and the color intensity of the obtained materials was measured by registering the absorbance at 433 nm of 0.5% (*w*/*v*) CR-CMC@ aqueous solutions, being this wavelength the maximum of the absorption band of *o*-cresol red at neutral pH. In Table 3, the tested dye amounts and the corresponding absorbance value at 433 nm are reported. Considering that no significant difference was registered in color intensity with lowering the amount of dye, the minimum value, i.e., 0.05 mmol, was kept as the starting dye amount.

### 3.2. Physico-Chemical Characterisation of CR-CMC@

DSC analyses were performed on CR-CMC@ powder, obtained after the synthesis. Due to CMC hydrophilicity, the heating process was preceded by an isothermal heating step to remove the sorbed humidity. In Appendix A, the calorimetric profiles of CMC and CR-CMC@ before the isothermal heating step are reported as an example. After water removal, no significant transitions occurred in the investigated range, analyzing CMC powder before and after functionalization, as clearly displayed by the profiles in Appendix A.

The FT-IR spectra of CR, CMC and CR-CMC@ powder are reported in Appendix A as follows: the CMC and CR-CMC@ powder spectra were reasonably superimposable with different intensities of the absorption bands; upon comparison with the IR spectrum of CR, it was clear that this technique was not suitable to confirm a successful covalent binding of the indicator to the CMC. Even though some tiny shoulder variation could be observed between 1450 cm^−1^ and 902 cm^−1^, the loaded CR amount was too low for infrared detection.

On the other hand, ^1^H-NMR spectra allowed us to confirm the successfully covalent binding of the indicator to the CMC: in particular, analyzing the aromatic region in ^1^H-NMR spectra of CR, CMC and CR-CMC@, reported in Appendix A, the CR signals could be clearly found between 6.6 and 8.0 ppm. Because of the covalent binding between CR and CMC, the CR aromatic protons at 6.82, 7.07 and 7.22 ppm clustered at 7.20 ppm due to the effect of the carbonyl group nearby. For further confirmation, we acquired the ^1^H-NMR spectrum of a mixture of CR and CMC, reported in Appendix A, where the two molecules of interest were present but not bound one to the other; in this case, the aromatic signals of CR were perfectly identical to those registered for the pristine molecule without any signals shift. Hence, we could demonstrate that the synthetic pathway developed allowed us to covalently bind sulfonated molecules to CMC.

The optical behavior of Dye-CMC@ was investigated by UV-Vis spectroscopy. The Dye-CMC@ powders were solubilized in water at a fixed concentration of 0.5% (*w*/*v*) and UV-Vis spectra were registered in order to compare the optical behavior of the dye in an aqueous solution and after functionalization at different pH values. In Figure 3, the UV-Vis spectra for CR (~10 μM) and CR-CMC@ (0.5% *w*/*v*) in solution are displayed as follows: UV-Vis spectra and corresponding photographs after equilibration at acidic, basic and neutral pH. The UV-Vis absorption properties of CMC were also investigated. 

No significant differences in the optical behavior were detected in terms of maximum absorption wavelength and, assuming that the presence of CMC did not affect the molar absorption coefficient, the amount of CR contained in CR-CMC@t was estimated from the absorbance value at 433 nm at pH = 7 and turned out to be 0.002 mmol g^−1^. Unfortunately, we could not separate the contribution of CR covalently bound to CMC and it remained physically entrapped in the powder even after the washing procedure.

### 3.3. Investigation of Protonation Equilibria of CR, CMC and CR-CMC@ 

#### 3.3.1. CR System

CR can be classified as a sulfonphthalein. This class of molecules is characterized by the following three acidic groups: sulfonic, hydroquinonic and phenolic. Being the sulfonic group a strong acid, CR can be considered as a weak diprotic acid (Appendix A), which can therefore exist in solution in the following three different forms depending on the pH: the fully protonated species H_2_(CR), which forms orange-red solutions under the most acidic conditions, H(CR)^−^ (yellow) and CR^2−^ (purple-red) [29,30]. 

In the explored pH range (2.5–10.5), it was possible to calculate only the thermodynamic constant corresponding to the first protonation equilibrium (log*β*_1_ = log*K*_1_) as follows: CR^2−^ + H^+^ = H(CR)^−^. The protonation constant (Table 4) obtained by potentiometric and pH-dependent UV-Visible titrations is log*K*_1_ = 8.14(8) and 8.212(3), respectively. These results are consistent with data reported in the literature for the same system in similar experimental conditions [29,30,31,32]. The species distribution diagram for protonation equilibria of CR is shown in Appendix A. The optical properties of the two species present in the solution, CR^2−^ and H(CR)^−^, are clearly observable from the registered absorption spectra (Appendix A). The wavelengths of maximum absorption (λ_max_) of the monoprotonated and deprotonated species are 434 and 573 nm, respectively. In Appendix A it is also possible to identify an isosbestic point at λ = 483 nm, which well fits with the literature value for H(CR)^−^/CR^2−^ equilibrium [32]. Furthermore, the UV-Vis absorption spectrum at pH 2.3 shows a slight shoulder at about 520 nm, most likely corresponding to the absorption of the fully protonated form of cresol red, H_2_(CR) (theoretical λ_max_ = 518 nm [31,32]), which may be present in solution in a very low, but still qualitatively observable concentration.

#### 3.3.2. CMC System

At first, to qualitatively evaluate the acid-base behavior of CMC, its potentiometric titration curve was compared with that of an aqueous solution of HCl 0.005 M and *I* = 0.1 M (KCl), i.e., the medium in which CMC (and the other investigated systems) was dissolved and that served as a reference curve (Appendix A). The comparison shows a major difference in the acidic pH range attributable to the presence of carboxymethyl groups (-CH_2_COO^−^) and their acid-base reaction. Although the polymeric nature of CMC does not allow for rigorously processing the potentiometric data, it was possible to calculate an average “apparent” protonation constant, which takes into account the acidic properties of the carboxymethyl groups present in the polymer as a whole (Table 5). Indeed, CMC is a weak polyelectrolyte where the apparent protonation constant of COOH groups is a function of the degree of substitution, the degree of dissociation (determined by the pH of the medium) and the ionic strength [33,34]. At the employed experimental conditions, the calculated log*K* for -CH_2_COOH groups is 3.86(5). Lastly, the hydroxyl groups of the glucopyranose monomer units of CMC are very weak acids (p*K*_a_ > 12 in glucose [35]) and their protonation equilibria cannot be properly investigated in the explored pH range (2.5–10.5).

#### 3.3.3. CR-CMC@ System

The investigation of the acid-base properties of CR-CMC@ requires the characterization of both the protonation equilibria of carboxymethylcellulose and cresol red. In CR-CMC@, there are, in fact, the following two main types of protonable sites: (i) the free COOH groups belonging to CMC and which are not covalently bound to the dye, and (ii) the CR molecules anchored to CMC. 

The potentiometric data processing allowed us to determine only the constant related to the free carboxyl groups. The experimental conditions were the same as employed for CMC (see Section 3.3.2), and the obtained average protonation constant (log*K* = 3.94(1)) varies only slightly from that of pure CMC, probably due to the different number of free carboxymethyl groups involved in the proton exchange equilibria [33]. On the other hand, it was not possible to determine the protonation constants of CR by potentiometry. The concentration of CR molecules in the titrated solutions, calculated on the basis of the dyeing capacity of CMC (0.002 mmol g^−1^), was in fact much lower than the detection limit of the potentiometric technique. The viscosity of the CR-CMC@ system prevented the use of more concentrated samples. Nevertheless, UV-Vis absorption spectrophotometry allows for the analysis of more diluted solutions and this technique was employed for the determination of CR protonation constants in the CR-CMC@ system (Appendix A). As in the case of CR (Section 3.3.1), in the explored pH range, only the first protonation constant was determined. The obtained value of log*K*_1_ is 8.839(6), fairly different from the calculated constant of pure CR (8.212(3), Table 4). The cresol red indicator, when anchored to carboxymethylcellulose, is therefore a slightly weaker acid than its pure form. At this point, we cannot exclude that the different protonation constant of CR in the conjugated system CR-CMC@ arises from a matrix effect of the polymeric environment in which the dye is found. To verify it, pure CR (with a final concentration [CR] = 1.1 × 10^−5^ M) was dissolved in a solution of CMC 0.5% *w*/*v* and *I* = 0.1 M (KCl), and the protonation constant (log*K*_1_) of CR was then determined by means of pH-dependent UV-visible titrations. The obtained log*K*_1_ is 8.209(5) and practically coincides with the value calculated by the spectrophotometric method for pure CR dissolved in water (8.212(3), Table 4). This result suggests the absence of a matrix effect.

### 3.4. Optimization of CR-CMC@ Drop Casting on Pure Cellulose

CR-CMC@ was then used as a reactive material to prepare colorimetric sensors for the detection of weak acids or bases in the vapor phase via drop-casting deposition [36,37,38,39]. In a preliminary investigation, several types of paper-based supports had been tested, ranging from filter paper, food-grade paper and paperboards, but the final choice fell on pure cellulose since this support allowed a complete absorption of the deposited drop and did not interfere in the detection of the analytes. 

Once selected as the support, the aqueous solution composition and drop volume were optimized by full factorial design 2^3^ as follows: in more detail, CR-CMC@ concentration in a range between 5 and 7.5% (*w*/*v*) was studied, having in mind that a higher CR-CMC@ concentration led to more intense sensor color but a slower detection rate, having a higher number of receptors to be protonated before showing complete color conversion. The amount of glycerol added as an additive was then investigated since it is well-known in literature the ability of this compound in helping CMC solubilization and in reducing CMC aqueous solution density as follows: for this reason, its quantity was not investigated varying the absolute value but its ratio in respect to CR-CMC@ from 0.5 to 1 glycerol/CR-CMC@. Finally, drop volume was investigated in a range between 10 and 20 μL to evaluate the effect of sensor dimension on detection kinetics.

Once defined variable levels, the plan of experiments and the training sensors, reported in Appendix A, labeled from 1 to 8 and the test one (n°9), were prepared and the detection kinetic towards acidic volatile analytes originated from 100 mL AcOH 0.01 M in a sealed box (V = 1.75 L) was measured by acquiring sensor pictures for 5 h, which are reported in Figure 4.

The pictures reported in Figure 4 clearly highlighted that the main differences between the prepared sensors are color intensity and detection rate as follows: as for the first feature, sensors with higher CR-CMC@ content, labeled with odd numbers, had a much more intense coloration than those with lower CR-CMC@ content (even numbers). Moving to the detection rate, it was more difficult to relate this property to one of the variables under investigation directly by looking at the pictures. To establish a mathematical relationship between these two features and the sensors compositions, two responses were calculated from the following sensors pictures: (i) the time required for complete color transition, identified as an H value that differs less than 10% from the final H value and (ii) the color saturation at the acquisition time defined above, both values acquired from HSL space color. In this context, L, lightness, is not informative. Being our aim to simultaneously maximize color intensity and minimize detection kinetic, two different models were developed, one per each response and validated by comparing the experimental and predicted response values for the center point, labeled as 9 in Figure 4. For brevity’s sake, the evolution of H and S values for each sample and the details of both the models are reported in Appendix A. 

After that, a Pareto optimal-front was built to select the best compromise between the following two responses: to build this plot, the two responses were plotted together in a scatter plot and the points were located in the graph according to their responses value, as showed in Figure 5 [40,41,42]. Once defined the targets for the investigation, in our case minimization of detection rate on the *x*-axis and maximization of color intensity on the *y*-axis, non-dominated points were highlighted as the points that represented the better compromises, labeled in red. Among these points, a subjective selection of the best composition could be made depending on the relative importance of the following two responses measured: for instance, in our case, samples 5 and 6 were discarded because the color intensity was too low, even if the detection was very fast and, similarly, sample 3 required too much time to show a complete color conversion. Therefore, sample 1 represented the best compromise ensuring a rapid and glaring color transition in presence of volatile weak acids.

To conclude, the optimization of CR-CMC@ drop-casting deposition on pure cellulose by the design of experiments allowed us to identify the best conditions for sensor preparation, summarized in Table 6.

### 3.5. Physico-Chemical Characterisation of CR-CMC@/Cell Sensor

The FT-IR spectra of CR-CMC@, glycerol, cellulose and CR-CMC@/Cell sensor are reported in Figure 6. From IR spectra comparison, it was clear that the CR-CMC@/Cell sensor spectrum follows the pattern of the CR-CMC@ with strong enhancement of some bands due to the presence of glycerol, e.g., at 3260, 2920 and 2874 cm^−1^, respectively. The cellulose contribution was boundary and can be slightly observed in the fingerprint zone of the spectrum, typically at 1020 cm^−1^.

In Figure 7, SEM images at increasing magnifications, focused on the CR-CMC@ drop border, are reported. In the image at a lower magnitude (Figure 7a), we could observe that the fibers are not strictly assembled and present empty spaces in the lattice and random disposition. The images at a higher magnitude, displayed in Figure 7b,c, allowed us to evaluate the interaction between support and drop and drop’s homogeneity as follows: the drop was sorbed by cellulose material, partially penetrated within the support fibers and this interaction resulted in a homogeneous sensor’s surface, without cracks.

### 3.6. A “Proof-of-Concept” of CR-CMC@/Cell Sensors Application for Food Freshness Detection

Finally, a preliminary investigation into the applicability of CR-CMC@/Cell sensors in real cases was performed as follows: among the various fields of application of pH-sensitive devices, food freshness monitoring was selected as “proof-of-concept” for the following several reasons: first, this topic has gained ever-increasing importance in the last years together with the increasing concerns about food safety and waste; secondly, the biocompatible nature of the materials employed made this device eligible for application in the field of food packaging and, last but not least, the reactive dye here bound to CMC, *o*-cresol red, has been already proposed as a food freshness indicator in plastic-based prototypes [9,12,13,26,27,28], thanks to its ability to clearly detect the acidic by-products released in the first part of protein food degradation. 

For this reason, we tested the applicability of CR-CMC@/Cell sensors for the detection of weak acids released by different cuts of chicken and turkey meat, ranging from breast slices to minced meats, during chilled storage in a domestic fridge (4 °C) for 5 days. In Figure 8, the color evolution of CR-CMC@/Cell sensors exposed in a sealed box (V = 1.75 L) over 300 g of different cuts of chicken and turkey meat is reported. As could be expected, the sensors showed a glaring transition from violet to yellow after the reaction with acidic by-products released by the tested food during spoilage. We could also observe that the color conversion seemed a bit faster in the presence of minced meat rather than breast slices; this could probably be due to the higher surface of meat exposed to air in the case of minced samples and, more generally, to the higher probability of contamination during food processing. 

Basing on this “proof-of-concept”, colorimetric sensors obtained by drop-casting deposition of functionalized CMC on cellulose or other cellulose-based supports could be considered eligible candidates for several applications, among which food freshness monitoring might be a starting point, as this topic is widely discussed in the literature.

## 4. Conclusions

An interesting and versatile approach for CMC covalent functionalization was proposed; the synthetic approach was tested on several sulfonated dyes, always obtaining high yields and a satisfying degree of functionalization, as given by the intense coloration of the final Dye-CMC@ powder. Furthermore, both the reaction and the following workup required reasonably short times and the final material was obtained in the form of ready-to-use dry powder. From the characterization, we confirmed the covalent binding of CR to CMC and we acquired valuable information on the thermal behavior of the powder, and a comparison between the acid-base properties of CMC, starting dye and functionalized CMC was made. Finally, an example of CR-CMC@ application in the field of colorimetric sensors was described, starting from the optimization of drop-casting deposition of the sensors by the design of experiments approach to an example of application on real samples. Among the countless applications of pH-sensitive sensors, food freshness detection was presented as a “proof of concept” due to the high interest encountered by these devices recently and the well-known applicability of the test dye in this field.

## 5. Patents

The synthetic pathway here presented was included in the PCT patent [43], then extended to Europe and the USA [44,45], covering the covalent anchoring of sulfonated dyes on CMC and other materials, and their application as freshness sensors for food freshness monitoring.

## Figures and Tables

**Figure 1 polymers-14-04061-f001:**
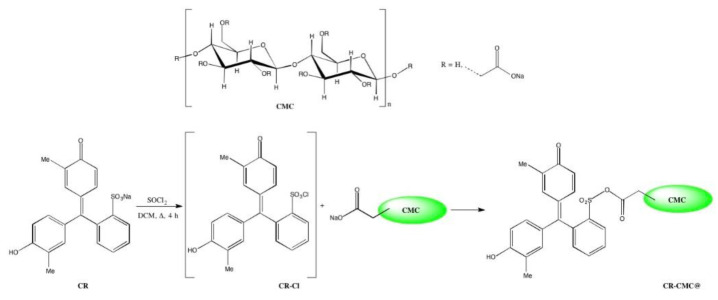
Reaction scheme for covalent anchoring of *o*-cresol red to CMC.

**Figure 2 polymers-14-04061-f002:**
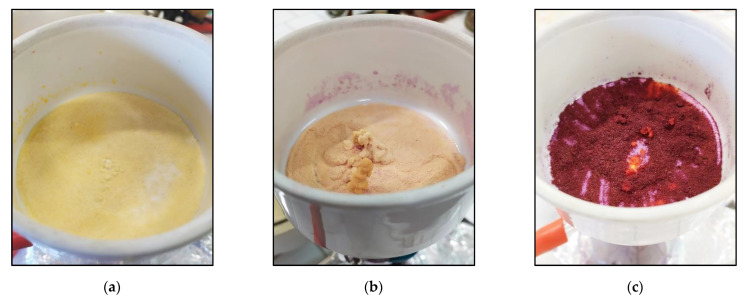
CR-CMC@ precipitated after synthesis using DMF (**a**), DCM (**b**) and toluene (**c**) as reaction solvent.

**Figure 3 polymers-14-04061-f003:**
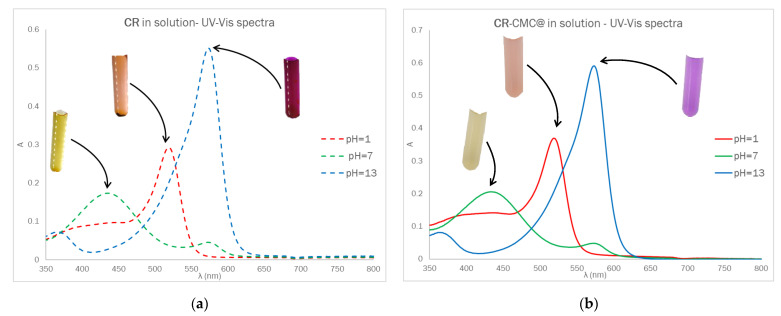
UV-Vis spectra and corresponding photographs of a ~10 μM CR solution (**a**) and 0.5% (*w*/*v*) CR-CMC@ solution (**b**).

**Figure 4 polymers-14-04061-f004:**
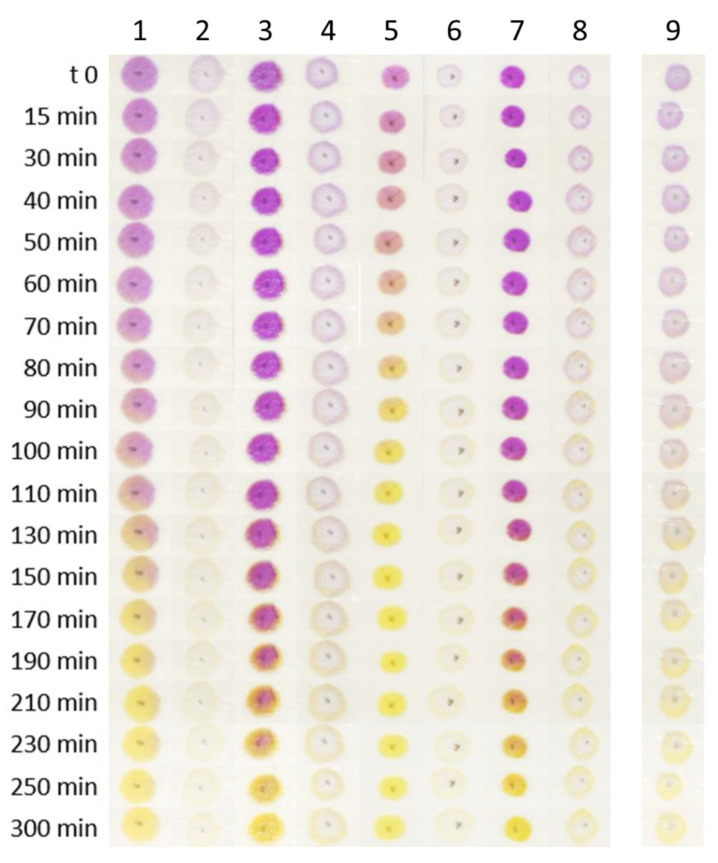
Pictures acquired during detection of acidic volatile analytes originated from 100 mL AcOH 0.01 M in a sealed box (V = 1.75 L) for each of the 8 + 1 sensors of the experimental plan.

**Figure 5 polymers-14-04061-f005:**
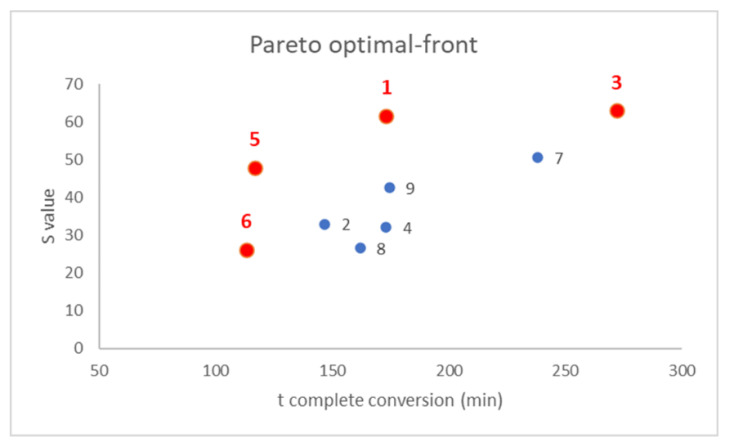
Pareto optimal-front reporting the two responses, time required for complete conversion (*x*-axis) and S value (*y*-axis) for the prepared sensors, labeled from 1 to 9 as reported in Appendix A.

**Figure 6 polymers-14-04061-f006:**
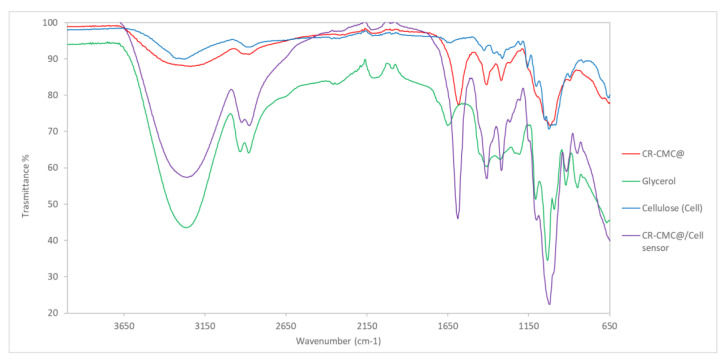
FT-IR spectra of CR-CMC@ (red), glycerol (green), cellulose (light blue) and CR-CMC@/Cell sensor (violet).

**Figure 7 polymers-14-04061-f007:**
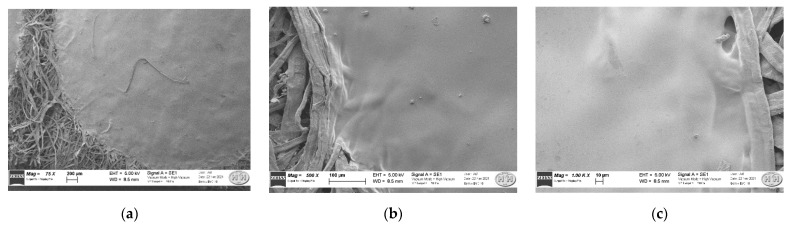
SEM images of CR-CMC@/Cell sensor at 75× (**a**), 500× (**b**) and 1000× (**c**).

**Figure 8 polymers-14-04061-f008:**
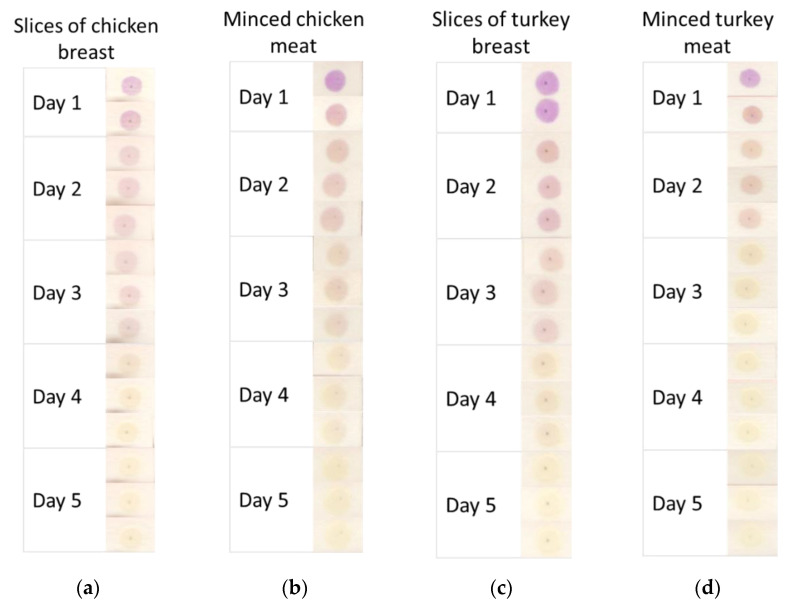
Color evolution of CR-CMC@/Cell sensors over slices of chicken breast (**a**), minced chicken meat (**b**), slices of turkey breast (**c**) and minced turkey meat (**d**) during chilled storage at 4 °C for 5 days.

**Table 1 polymers-14-04061-t001:** Levels definition for CR-CMC@ concentration, glycerol/CR-CMC@ ratio and drop volume.

Variable	Lower Level (-)	Upper Level (+)
CR-CMC@ concentration (% *w*/*v* H20)	5%	7.5%
Glycerol/CR-CMC@ ratio	0.5	1
Drop volume (µL)	10 µL	20 µL

**Table 2 polymers-14-04061-t002:** Average precipitation yields for CR-CMC@ synthesized in different reaction solvents.

Reaction Solvent	Work-Up	Yield%
DMF	--	87(4)%
DCM	--	97(2)%
Toluene	--	92(1)%
Toluene	Evaporation by half	98(1)%

**Table 3 polymers-14-04061-t003:** Dye amounts tested for CMC functionalization and A value at 433 nm registered for 0.5% CR-CMC@ aqueous solutions.

Dye Amount (mmol)	A (433 nm)
0.3	s0.221
0.2	0.208
0.1	0.212
0.05	0.206

**Table 4 polymers-14-04061-t004:** Protonation constant referring to the cresol red protonation step: CR^2−^ + H^+^ = H(CR)^−^ at *T* = 298 K and *I* = 0.1 M (KCl). Values in parentheses are standard deviations on the last significant figure.

System	log*K*_1_	Experimental Method
CR	8.14(8)	Potentiometry
	8.212(3)	UV-Vis spectrophotometry
CR-CMC@	8.839(6)	UV-Vis spectrophotometry

**Table 5 polymers-14-04061-t005:** Average apparent protonation constant referring to the carboxymethyl groups present in CMC at *T* = 298 K and *I* = 0.1 M (KCl). Values in parentheses are standard deviations on the last significant figure.

System	log*K*_1_	Experimental Method
CMC	3.86(5)	Potentiometry
CR-CMC@	3.94(1)	Potentiometry

**Table 6 polymers-14-04061-t006:** Optimized conditions for CR-CMC@ sensors preparation by drop-casting deposition on pure cellulose.

Variable	Optimized Value
CR-CMC@ concentration (% *w*/*v* H20)	7.5%
Glycerol/CR-CMC@ ratio	1
Drop volume (µL)	20 µL

## Data Availability

Not applicable.

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
