# Peer review of "Quick and Easy Covalent Grafting of Sulfonated Dyes to CMC: From Synthesis to Colorimetric Sensing Applications"

_polymers, 2022, doi:10.3390/polym14194061_

Round 1

Reviewer 1 Report

The authors presented a sensor for pH based on the covalent anchoring of a pH sensitive dye (o-cresol red, CR) to carboxymethyl cellulose (CMC).

The main issue is that the covalent grafting of CR to CMC was not supported by any experimental data. Thus, the title does not reflect the results. Consequently, one can ask if the results about the pH responsiveness comes from physically bound CR to CMC. This is an important point that must be clarified by additional experiments. Also, the authors should clearly state what is the novelty of CR-CMC in comparison to their recent publication about CR-EVOH based pH sensors.   

L37 – “Cellulose offers a wide variety of natural macromolecules” – It does not make sense. Please revise

L68-73 – This sentence is too long and confusing. Try to write short sentences. It is easier to follow the ideas.

L81 – Please provide the DS and the molecular weight.

L82 – Please provide the source and the molecular weight of pure cellulose.

Materials and methods - Why was the purity informed only for potassium hydrogen phthalate? Please provide the purity for all reactants.

L89 - plastic polymers???

L90-91 – “Here after the general procedure is reported while, in the following section, the procedure optimisation is discussed.” This sentence is badly formulated.

L91-93 – The sentence is confusing.

L95 – How much of CR was added to SOCl2?

L121 – were acquired

Fig. 1 – Please define Z

L214-215 - Although the authors state that, it is a well reported reaction they should briefly explain the reaction mechanism.

L261-262 – I agree with the authors that FTIR is not suitable to confirm a successful covalent binding of the indicator to the CMC. Please move this figure to the Supplementary Material.

The DSC curves gave no clear evidence that CR was successfully bound to the CMC.

The only technique that can provide clear evidence for the covalent binding of CR to CMC is NMR. The authors should run  1H NMR spectra for CR, CMC and CR-CMC@ samples. The signal can be integrated and the DS can be calculated. According to Figure 1, changes related to C6 signal are expected. Then the authors can affirm how much of CR was chemically bound to CMC chains.

Figure 4b – How sure can one be that the electronic spectra is not from leached CR molecules, whcih were just physically adsorbed on CMC?

Actually, the main issue here is that it is not clear how much of the pH response comes from CR molecules physically bound to CMC and how much comes from CR molecules chemically bound to CMC.  There is no problem, if the CR molecules are just physically bound to the CMC, but it must be  quantified and the title should be changed. As it is, it is very confusing. The authors should consider quantifying the amount of  sulfur by ICP-OES to calculate the amount of physically incorporated CR. 

I recommend major revision and additional experiments to understand the CR-CMC@ system, which is used as sensor. 

Reviewer 2 Report

The manuscript entitled “Quick and easy covalent grafting of sulfonated dyes to CMC:

from synthesis to colorimetric sensing applications”. Some issues to be addressed which will improve the quality of manuscript. Therefore, I recommend this work could be published after the major revision

1.      Should author write down the novelty of this paper?

2.      The English composition requires many improvements. The authors should proofread the manuscript carefully to minimize grammatical errors.

3.      All the references mentioned in the paper should be cited in the text or vice-versa.

4.      This research topic has been widely studied, and many studies have been performed. The author, please add a comparative table for the reader's clear understanding.

5.       Please increase the font size within all figures so the reader can see them well.

6.      Please write the conclusion short and up to a point.

7.       The author must indicate the FTIR peak value.

8.      The author should improve the color quality of Fig. 3.

Round 2

Reviewer 1 Report

The authors did a good job providing the 1H NMR spectra to support the chemical modification of CMC. The study has similarities to a previous work of the same group involving similar modification of EVOH for sensors.  However, CMC is a low cost cellulose derivative, which  offers many applications, which require non-toxic materials. The suggested modifications were properly done and I recommend for publication.

Reviewer 2 Report

The author did all necessary changes in the final version of the manuscript, I recommend accepting it in the present form.